# Density estimates of monarch butterflies overwintering in central Mexico

Wayne E. Thogmartin[1], Jay E. Diffendorfer[2], Laura López-Hoffman[3],
Karen Oberhauser[4], John Pleasants[5], Brice X. Semmens[6],
Darius Semmens[2], Orley R. Taylor[7] and Ruscena Wiederholt[8]

[1] Upper Midwest Environmental Sciences Center, US Geological Survey, La Crosse, WI, USA
[2] Geosciences and Environmental Change Science Center, US Geological Survey, Lakewood, CO, USA
[3] School of Natural Resources and the Environment and Udall Center for Studies in Public Policy, University of Arizona, Tucson, AZ, USA
[4] Department of Fisheries, Wildlife and Conservation Biology, University of Minnesota, St. Paul, MN, USA
[5] Department of Ecology, Evolution, and Organismal Biology, Iowa State University, Ames, IA, USA
[6] Scripps Institution of Oceanography, University of California, San Diego, La Jolla, CA, USA
[7] Department of Ecology and Evolutionary Biology, University of Kansas, Lawrence, KS, USA
[8] Everglades Foundation, Palmetto Bay, FL, USA

Corresponding author
Wayne E. Thogmartin,
wthogmartin@usgs.gov

## ABSTRACT

Given the rapid population decline and recent petition for listing of the monarch butterfly (*Danaus plexippus* L.) under the Endangered Species Act, an accurate estimate of the Eastern, migratory population size is needed. Because of difficulty in counting individual monarchs, the number of hectares occupied by monarchs in the overwintering area is commonly used as a proxy for population size, which is then multiplied by the density of individuals per hectare to estimate population size. There is, however, considerable variation in published estimates of overwintering density, ranging from 6.9–60.9 million ha$^{-1}$. We develop a probability distribution for overwinter density of monarch butterflies from six published density estimates. The mean density among the mixture of the six published estimates was $\sim$27.9 million butterflies ha$^{-1}$ (95% CI [2.4–80.7] million ha$^{-1}$); the mixture distribution is approximately log-normal, and as such is better represented by the median (21.1 million butterflies ha$^{-1}$). Based upon assumptions regarding the number of milkweed needed to support monarchs, the amount of milkweed (*Asclepias* spp.) lost (0.86 billion stems) in the northern US plus the amount of milkweed remaining (1.34 billion stems), we estimate >1.8 billion stems is needed to return monarchs to an average population size of 6 ha. Considerable uncertainty exists in this required amount of milkweed because of the considerable uncertainty occurring in overwinter density estimates. Nevertheless, the estimate is on the same order as other published estimates. The studies included in our synthesis differ substantially by year, location, method, and measures of precision. A better understanding of the factors influencing overwintering density across space and time would be valuable for increasing the precision of conservation recommendations.

## INTRODUCTION

"I can see no other escape from this dilemma (lest our true aim be lost forever) than that some of us should venture to embark on a synthesis of facts and theories, albeit with second hand and incomplete knowledge of some of them—and at the risk of making fools of ourselves." (*Schrödinger, 1944*: 1)

Monarch butterflies overwintering in the high-elevation Oyamel fir (*Abies religiosa*) forests of central Mexico form spectacular aggregations thought to number in the millions of individuals per hectare (*Urquhart & Urquhart, 1976*; *Brower, 1977*). The cool temperatures of these high-elevation sites allow monarchs to slow their metabolism, conserving lipid reserves for the approximately 5-month wintering period. Clustering in densely packed colonies on the lower branches of Oyamel fir trees also minimizes mortality during cold and rainy winter nights (*Anderson & Brower, 1996*; *Brower et al., 2009*; *Williams & Brower, 2015*) and increases humidity, thus reducing evaporation and desiccation as the dry season advances (*Brower et al., 2008*). In early spring, migration of monarchs over much of eastern North America resumes from this location, a multi-generational migratory phenomenon seen in few other insects.

Since winter 1994–1995, World Wildlife Fund-Mexico (WWF) in collaboration with the Mexican Secretariat of Environment and Natural Resources (Secretaría de Medio Ambiente y Recursos Naturales; SEMARNAT), the National Commission for Protected Areas (Comisión Nacional de Áreas Naturales Protegidas; CONANP), and the Monarch Butterfly Biosphere Reserve (MBBR) have monitored the overwintering population. The winter monitoring consists of estimating the area over which these densely packed colonies occur (*Calvert & Brower, 1986*; *Garcia-Serrano, Reye & Alvarez, 2004*; *Slayback et al., 2007*; *Vidal, López-García & Rendón-Salinas, 2014*; *Vidal & Rendón-Salinas, 2014*; *Rendón-Salinas & Tavera-Alonso, 2014*). Occupied trees are mapped in each colony and the perimeter of the colony is measured. The enclosed area is then calculated in hectares occupied and used as an index of population size.

This monitoring of the Eastern population of monarch butterflies (*Danaus plexippus*) in North America suggests large declines in the wintering population size over the last decade and a half (*Semmens et al., 2016*). The largest population size recorded since monitoring began in the early 1990s was 18.19 ha in winter 1996–1997. Since this peak in abundance, monitoring suggests that the population has declined by over 90% (*Brower et al., 2012*; *Vidal & Rendón-Salinas, 2014*; *Rendón-Salinas & Tavera-Alonso, 2014*), to a record low of 0.67 ha in winter 2013–2014 (*Rendón-Salinas & Tavera-Alonso, 2014*). These declines in abundance are believed to be due, in large part, to declines in habitat availability in the breeding range of the north-central United States, principally through loss of common milkweed (*Asclepias syriaca*) in agricultural crops (*Pleasants & Oberhauser, 2013*; *Pleasants, 2015*; *Pleasants, 2017*), as well as forest degradation in the Mexican overwintering habitat (*Brower et al., 2016*).

In 2014, due to concerns over these overwintering population declines, the US Fish and Wildlife Service was petitioned to list monarchs as a threatened species under the

Endangered Species Act (*Center for Biological Diversity et al., 2014*; docket number FWS-R3-ES-2014-0056). The agency subsequently initiated a status review to determine whether listing for the entire species was warranted. The White House announced a strategic goal of increasing the eastern population of the monarch butterfly to 225 million butterflies by 2020 (*Pollinator Health Task Force, 2015*). This 225 million butterfly goal was motivated in part by the premise that 225 million butterflies equated to 6 ha of habitat occupied by monarch butterflies in overwintering sites (*Pollinator Health Task Force, 2015*), or 37.5 million butterflies ha$^{-1}$. The magnitude of this target for the Eastern migratory population of monarch butterflies has important implications for the estimated level of restoration effort (i.e., increasing milkweeds) needed to sustain the population in eastern North America. Therefore, accurate determination of the overwintering population size, not just the area over which it occurs in winter, is a critical step in determining the magnitude of the conservation challenge.

To translate from colony extent to population numbers, estimates of the area occupied by overwintering aggregations must be multiplied by an estimate of density (monarchs/unit area). Current understanding of the overwintering densities of monarch butterflies in these aggregations comes from a handful of published sources, principally *Brower et al. (1977)*, *Brower et al. (2004)* and *Calvert (2004)*. *Brower et al. (1977)* and *Tuskes & Brower (1978)* used density estimates from capture-mark-recapture for California overwintering colonies for a rough estimate of abundance in Mexico. They multiplied the density estimate for Santa Cruz, California, which was 95,000 butterflies ha$^{-1}$, by 15 to account for the difference in area covered by Mexican and Californian colonies, and again by 10 to account for their suggestion that California colonies were 10% of the density of the Mexican colony. Their suggestion of 14.25 million monarch butterflies occupying 1.5 hectares in one location in Mexico was deemed "a conservative estimate" of 9.5 million monarch butterflies ha$^{-1}$. It was not until nearly a quarter-century later that attempts at calculating density using on-site measurements made in Mexico were published. *Calvert (2004)* used two approaches with capture-mark-recapture data to estimate population densities of 21 to 100 million monarchs ha$^{-1}$, with higher densities occurring later in the season when the colony had contracted. At a different colony, *Calvert (2004)* measured monarch density on a sub-sample of tree branches and trunks and generated an estimate of 10.3 million monarchs ha$^{-1}$. *Brower et al. (2004)* took a different tack, extrapolating the density of monarchs killed during a winter storm (in January 2002) at two sites, and obtained estimated densities of 53 and 73 million monarchs ha$^{-1}$ for the two sites, for a mean estimate of 65 million monarchs ha$^{-1}$. This mean estimate was subsequently revised down to 50 million monarchs ha$^{-1}$ (*Slayback et al., 2007*) to be more conservative (L Brower, pers. comm., 2016). Obviously, considerable variation exists in the estimates of overwintering densities, which has important policy ramifications for their use by groups working together to chart a strategy for protecting monarchs, including the US Fish and Wildlife Service, other partner agencies in the US, Mexico and Canada, and non-governmental actors.

Accurate estimation of the density of monarch butterfly populations overwintering in Mexico provides critical information for determining the abundance of the Eastern

**Table 1 Densities (in millions ha$^{-1}$) of monarch butterflies overwintering in central Mexico, by method and source with the estimated standard deviation.**

| Method | Publication | Date of study | Density | SD |
|---|---|---|---|---|
| Petersen capture-mark-recapture | *Calvert (2004)* | Late-Dec 1985 | 6.9 | 1.2 |
| Jolly-Seber capture-mark-recapture | *Calvert (2004)* | Early-Jan 1986 | 33.8 | 1.3 |
| Storm mortality-based (Zapatero) | *Brower et al. (2004)* | Mid-Jan 2002 | 18.4 | 20.1 |
| Storm mortality-based (Conejos) | *Brower et al. (2004)* | Mid-Jan 2002 | 15.9 | 24.4 |
| Petersen capture-mark-recapture | *Calvert (2004)* | Late-Jan 1986 | 60.9 | 1.2 |
| Branch extrapolation | *Calvert (2004)* | ca. Early-Feb 1979 | 10.3 | 2.1 |

migratory population. Uncertainties in these density estimates may arise from many different sources such as natural variability in the monarch's response to environmental conditions, including their own population numbers, and variability in environmental stressors over time and space (*Williams & Brower, 2016*). Incomplete knowledge regarding a situation or variable, often occurring as a result of measurement (or observation) error, unstated assumptions or extrapolations, also contributes uncertainty in density estimates. Methods have been developed for estimating variables when faced with stochastic variation and incomplete knowledge (*McLachlan & Peel, 2000*; *Zadeh, 2002*; *Pearson, 2011*). Based on the form of available information, probability theory can be used to incorporate parameter uncertainty and variability into an expected value distribution (*Shapiro, 2009*). The expected value distribution describes the distribution around the expected value, or the weighted average of all possible outcomes. Here we used finite mixture distribution modeling to derive the expected distribution of monarch overwintering density from the estimates of *Calvert (2004)* and *Brower et al. (2004)*. We then applied the median estimated abundance from that distribution to a corrected time series of overwinter abundance (*Semmens et al., 2016*) to understand the magnitude of population change since systematic monitoring for wintering monarchs began in the early 1990s. We also propose an environmental correlate to changing within-season density. Additionally, we used our estimated density to estimate the amount of milkweed needed to sustain the 6-ha goal population.

## METHODS

We calculated the fuzzy random variable for monarch density using two general steps. First, for each of six available estimates of density, we calculated the uncertainty around the estimated central tendency (estimated mean or reported value) and then modeled this using a lognormal distribution (except in one case, where extreme values required an extreme value distribution). Patterns in species abundance are often lognormal (*Sugihara, 1980*; *Limpert, Stahel & Abbt, 2001*). This resulted in five lognormal distributions and one extreme value distribution, each centered on the original point estimate of density (Table 1). We then combined the six distributions to estimate a new, combined distribution of density incorporating putative levels of uncertainty in the underlying reported estimates. The six available estimates of density come from two sources, four estimates from *Calvert (2004)* and two from *Brower et al. (2004)*.

*Calvert (2004)* used the Petersen and Jolly-Seber capture-mark-recapture methods for calculating mean and 95% confidence interval estimates of overwinter density from the capture records of tagged butterflies in late December 1985 and mid-January 1986. The reported confidence intervals were asymmetrical. We approximated the population standard deviations, on the natural log scale, from these confidence intervals according to $\tilde{\sigma} = \left( \frac{\log(mean) - \log(lower\ 95\%\ limit)}{2.11} + \frac{\log(upper\ 95\%\ limit) - \log(mean)}{2.11} \right)/2$, where 2.11 is the $t$ critical value under the assumption of a small sample size. Using the reported mean and estimated standard deviations, we fit both gamma and lognormal distributions (*Wilks, 2006*); the lognormal distribution fit the published confidence intervals best (matched most closely) and was used in the analysis.

*Calvert (2004)* also estimated overwinter density from a sample of monarch butterflies collected from branches and tree trunks in 1977. He used 12 branches of varying size to regress monarch abundance against branch size, and then used data on tree structure to estimate the average size and number of branches for trees of different sizes. He then measured the number of monarchs on 17 different tree trunks. He summed "crown monarchs" (on branches) and "trunk monarchs" to obtain an estimate of monarchs per tree. This branch-based estimate of density was reported without a measure of associated variance. Therefore, we treated Calvert's branch-based estimate as a Fermi approximation (*Machtans & Thogmartin, 2014*), a rough estimate for a difficult-to-estimate quantity, and inferred the variance to be a function of the number of parameters in the branch and trunk calculations. We assumed this estimate was correct within a factor of two; this range gives this estimate similar precision as the other January estimates of density. Thus, given this assumption, we calculated the upper limit in the crown and trunk estimates as $2^{\sqrt{n}} \times$ mean of the crown and trunk estimate, respectively, where $n$ equaled the number of parameters used in the calculation of the estimate. The lower limit was similarly calculated but with the inverse of $2^{\sqrt{n}}$ as the multiplier. Based on our reading of *Calvert (2004)*, we surmised there were five parameters in the crown estimate (diameter at breast height, crown mass, branches per crown, tree density, monarch weight) and three in the trunk estimate (surface area of a column, monarch sample density, tree density). Once the lower and upper limits in the estimates were established, the standard deviation was estimated as above for Calvert capture-mark-recapture data.

*Brower et al. (2004)* reported estimated densities from two colonies (Zapatero and Conejos) in mid-January 2002, and thereafter assumed a midpoint as the nominal mean density of colonies. We were provided the original data used in extrapolating storm mortality observations to hectare-scale density estimates. Details of data collection are provided by *Brower et al. (2004)* but, briefly, the data comprise of counts made of dead and moribund individuals observed in 29 0.2 m × 0.2 m plots in each of the two colonies. The mortality data are highly skewed, particularly those data from the Conejos colony (Appendix S1); for example, >1,000 dead and moribund monarch butterflies were counted in a single 0.04 m$^2$ plot. For the Zapatero colony, we fit a lognormal distribution to the observed counts, whereas for the highly skewed Conejos colony we fit a generalized extreme value distribution. The generalized extreme value distribution is characterized by mean $E[X] = \zeta - \beta[1 - \Gamma(1 - \kappa)]/\kappa$ and variance $\text{Var}[X] = \beta^2 \left( \Gamma[1 - 2\kappa] - \Gamma^2[1 - \kappa] \right)/\kappa^2$

(*Wilks, 2006*: 87), where $\kappa$ is a shape parameter, $\zeta$ is a location or shift parameter, and $\beta$ is a scale parameter. Because these distributions provide the expected count for a $0.04\ m^2$ area, we then extrapolated this distribution to the hectare scale (by multiplying by 250,000) to make them commensurate in scale with the other published estimates of density.

The six density methods differed substantially in their reported means. Our estimates are based on measurements from three years (1979, 1985–1986, 2002). These estimates (aside from one) are presumptively drawn from lognormal distributions. Elementary probability theory cannot describe the distribution of the sum of lognormals (*Dufresne, 2004*). Thus, we relied on mixture distribution modeling. Absent any data on the precision of the different methods, we developed a mixed probability density function $g$ as an equal-weighted sum of $k$ component densities: $g(\chi|x, \mu, \sigma) = \pi_1 f(\chi|\mu_1, \sigma_1) + \cdots \pi_k f(\chi|\mu_k, \sigma_k)$, where $k$ is equal to the six distributions described above. We created this distribution by drawing $10^6$ samples randomly from each distribution and then combining the drawn samples. Measures of central tendency and 2.5% and 97.5% quantile estimates were derived from the resulting mixture distribution.

Temperature and humidity play an important role in monarch overwintering behavior and roost suitability (*Anderson & Brower, 1996*; *Brower et al., 2008*; *Brower et al., 2009*; *Brower et al., 2011*). For instance, freezing dew on the surface of exposed butterflies can lower their supercooling resistance, ostensibly because ice crystals invade the spiracles of the butterflies, providing nucleation centers in the supercooled body fluids, which then may freeze (*Anderson & Brower, 1996*). With the published density estimate ascribed to the reported day of year (either the midpoint or endpoint), we quantified the observation that monarch butterflies pack more tightly with decreasing winter temperature (*Brower et al., 2011*; *Vidal & Rendón-Salinas, 2014*) by regressing the observed pattern of density to daily mean temperature and daily mean dew point (for the closest location for which these data were available, Toluca, Mexico, for the period 1977–2014: US National Climatic Data Center (NCDC) Global Summary of the Day; https://www7.ncdc.noaa.gov/CDO/cdo, downloaded Tue Dec 22 09:42:55 EST 2015). Dew point is the temperature (varying according to atmospheric pressure and humidity) below which water condenses; dew point is often correlated with minimum temperature, especially when humidity is high.

Once we derived the relevant density estimate, we calculated total abundance of monarchs and the associated milkweed (*Asclepias* spp.) required to sustain them. *Nail, Stenoien & Oberhauser (2015)* used Monarch Larva Monitoring Program egg density and survival data to estimate the number of milkweed plants needed to produce an adult monarch migrating to Mexico. Their calculations (see their equation 2) resulted in an estimate of 28.5 milkweed stems per monarch necessary to produce one adult for the fall migration. With this estimate from *Nail, Stenoien & Oberhauser (2015)*, we translated the 6 ha overwintering goal for monarchs into numbers of milkweed stems, under average climatic conditions (*Pleasants, 2017*).

We also applied the derived density estimate to the time series of overwintering abundances to estimate potential change in monarch population size through time. *Semmens et al. (2016)* used a state-space formula for estimating a corrected time series of the areal estimate of the overwinter population size. This state-space formula enabled

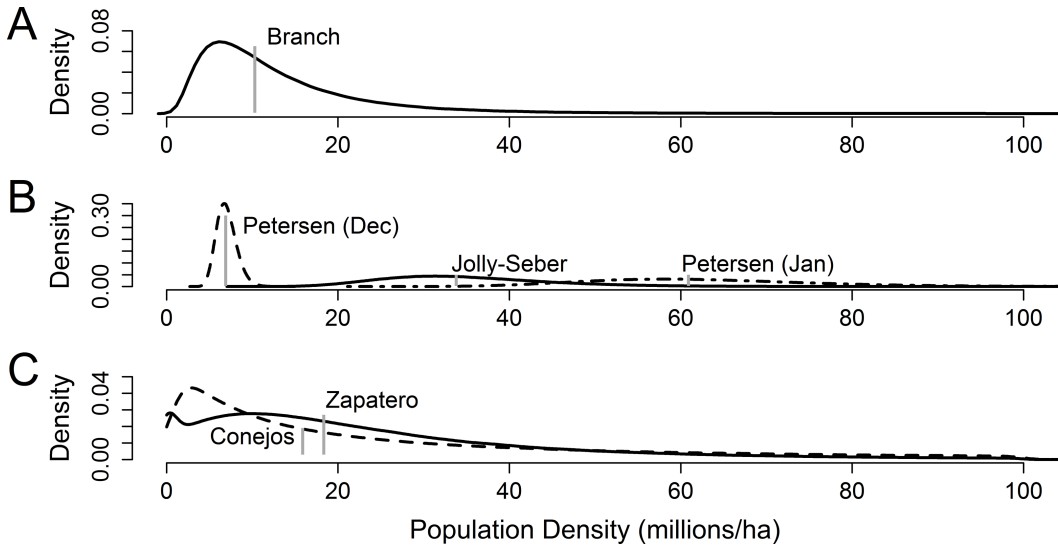

**Figure 1** **Presumptive distributions for estimates of overwinter monarch butterfly density in central Mexico.** (A) Presumptive distribution for structurally defined density used by *Calvert (2004)*. The vertical gray line is the published estimate. (B) Three capture-mark-recapture methods for estimating density as reported by *Calvert (2004)*, pertinent to the mid-January to early February sampling period. Vertical gray lines are the published estimate. (C) Presumptive distributions for storm-mortality method of density estimation used by *Brower et al. (2004)* from two colonies, Zapatero (solid line) and Conejos (dashed line); median densities are depicted in gray.

estimation of the underlying true state of the population corrected for observation noise. We multiplied the corrected estimate of areal overwinter population size by the median of the mixture distribution to calculate the annual monarch butterfly population abundance (given constant annual density).

## RESULTS

Reconstructed distributions matched the means reported by *Calvert (2004)*; the January capture-mark-recapture distributions showed high levels of uncertainty and overlapped each other, but were significantly larger than the January branch-based method (Fig. 1). The January branch-based method roughly coincided with the mean of the December capture-mark-recapture distribution. The *Brower et al. (2004)* storm-mortality approach and the *Calvert (2004)* January capture-mark-recapture methods reported results two to four times higher than the *Calvert (2004)* branch and December Petersen capture-mark-recapture methods (Fig. 1).

The mixture distribution was roughly lognormal in shape (Appendix S2) with a pronounced spike due largely to the branch-based and December Petersen capture-mark-recapture distributions (Fig. 2). The mean and median of this distribution were 27.8 and 21.1 million butterflies ha$^{-1}$, respectively (2.5% quantile = 2.4, 97.5% quantile = 80.7 million butterflies ha$^{-1}$).

Density regressed against temperature, dew point, and day of year most strongly supported a negative relationship between density and dew point (Table 2 and Fig. 3A).
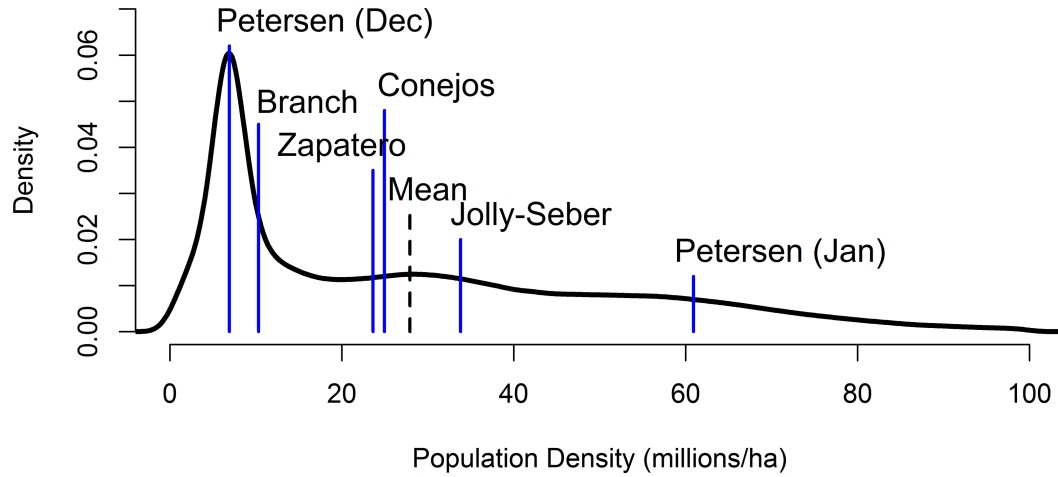

**Figure 2** A mixture distribution equally combining the individual distributions from the Jolly-Seber, December and January Petersen, Branch, and Brower storm mortality methods (means of the underlying distributions are denoted by the blue lines).

**Table 2** Density of monarch butterflies overwintering in central Mexico regressed against daily mean temperature, daily mean dew point, and day of year, where 12 Dec = 1, 20 Dec = 8, 1 Jan = 21, and 1 Feb = 52. $K$ is the number of parameters, AICc is the small-sample Akaike's Information Criterion, $\Delta$AICc is the difference between the best model and the focal model (AICc$_i$− minAICc), AICc $\omega$ is the model weight or conditional probability of the model relative to the other models in the model set and LL is the log-likelihood.

| Variables | $K$ | AICc | $\Delta$AICc | AICc $\omega$ | LL |
|---|---|---|---|---|---|
| Dew point | 3 | 63.90 | 0 | 0.87 | −22.95 |
| Day of year | 3 | 68.86 | 4.96 | 0.07 | −25.43 |
| Temperature | 3 | 69.40 | 5.50 | 0.06 | −25.70 |
| Temperature + Dew point | 4 | 92.39 | 28.49 | 0 | −22.20 |
| Dew point + Day of year | 4 | 93.44 | 29.54 | 0 | −22.72 |
| Day of year + Temperature | 4 | 98.57 | 34.67 | 0 | −25.28 |

When dew point is nearest 0 °C, monarch density is predicted to be greatest. Temperature and dew point in central Mexico are both lowest in mid-January (Fig. 3B).

Using the mixture distribution from the full set of density-estimation methods and assuming constant annual density during winter monitoring, the time series of overwinter population size suggested monarch butterflies may have numbered 310 million individuals in winter 1996–1997 and dropped to as low as 37 million in winter 2013–2014, nearly an order of magnitude difference (Table 3 and Fig. 4). The mean annual abundance over this 20-year period was 119 million butterflies (95% CI [69–212] million).

A 6-ha population goal using the median of the six-estimate mixture distribution equated to a mean of 127 million monarchs (6 ha × 21.1 million ha⁻¹). Assuming 28.5 milkweed stems are needed to produce a single monarch (*Nail, Stenoien & Oberhauser, 2015*), 127 million monarchs equaled ~3.62 billion stems (127 million monarchs × 28.5 milkweed/monarch).

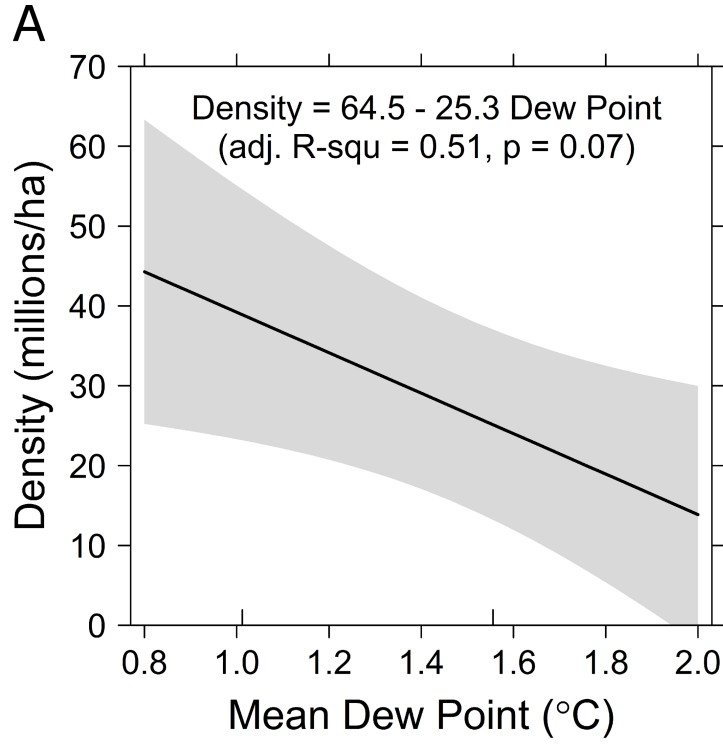

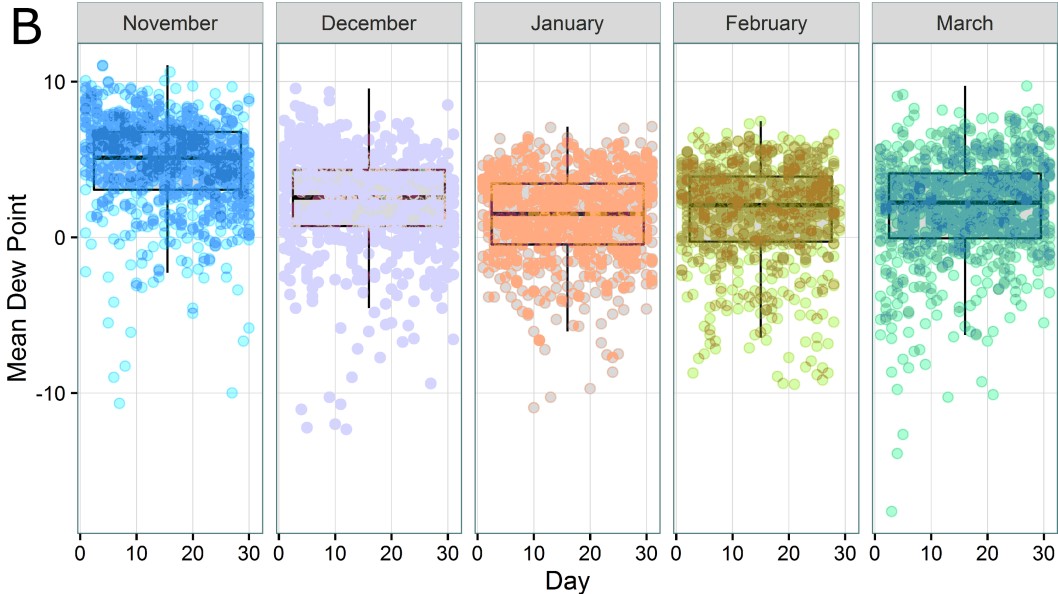

**Figure 3** **(A) Overwinter density of monarch butterflies as a function of mean daily dew point (°C). (B) Boxplots (median and 1st and 3rd quartiles, with 95% confidence interval whiskers) of observed daily dew points for each winter month over the period 1977–2015.**

**Table 3** Observed and fitted area of monarch butterflies overwintering in central Mexico with associated predicted population size (in millions of individuals) and 95% credible interval.

| Start year | Observed Ha | Fitted Ha[a] | 50% | 2.50% | 97.50% |
|---|---|---|---|---|---|
| 1993 | 6.23 | 6.79 | 155.4 | 72.5 | 342.6 |
| 1994 | 7.81 | 8.24 | 186.3 | 95.4 | 375.1 |
| 1995 | 12.61 | 12.00 | 267.1 | 133.0 | 529.8 |
| 1996 | 18.19 | 13.80 | 310.3 | 152.4 | 606.1 |
| 1997 | 5.77 | 6.77 | 155.8 | 81.8 | 320.8 |
| 1998 | 5.56 | 6.02 | 136.6 | 71.3 | 267.4 |
| 1999 | 8.97 | 6.95 | 151.2 | 102.7 | 227.1 |
| 2000 | 3.83 | 5.15 | 110.4 | 65.6 | 159.9 |
| 2001 | 9.36 | 7.00 | 151.6 | 103.6 | 235.9 |
| 2002 | 7.54 | 5.11 | 108.8 | 75.5 | 197.4 |
| 2003 | 11.12 | 5.32 | 112.8 | 78.0 | 229.4 |
| 2004 | 2.19 | 2.91 | 65.3 | 45.1 | 98.3 |
| 2005 | 5.91 | 4.15 | 90.3 | 62.6 | 148.2 |
| 2006 | 6.87 | 4.64 | 100.4 | 69.3 | 166.9 |
| 2007 | 4.61 | 4.18 | 90.9 | 62.9 | 140.1 |
| 2008 | 5.06 | 3.37 | 72.4 | 50.3 | 122.1 |
| 2009 | 1.92 | 2.52 | 56.4 | 37.8 | 82.8 |
| 2010 | 4.02 | 3.72 | 82.5 | 55.4 | 119.5 |
| 2011 | 2.89 | 3.18 | 71.6 | 46.3 | 103.5 |
| 2012 | 1.19 | 2.04 | 46.6 | 25.1 | 68.7 |
| 2013 | 0.67 | 1.59 | 37.1 | 16.9 | 54.2 |
| 2014 | 1.13 | 2.17 | 50.9 | 22.8 | 75.7 |

**Notes.**

[a] See *Semmens et al. (2016)* for details and credible intervals.

# DISCUSSION

Based upon reasonable assumptions regarding distributional form and characteristics of available published data, we suggest the preponderance of evidence supports a median overwintering density of monarchs of 21.1 million butterflies ha$^{-1}$ (2.5% quantile = 2.4 million butterflies ha$^{-1}$, 97.5% quantile = 80.7 million butterflies ha$^{-1}$). The few observations we have suggest monarch density changes with dew point, likely increasing in density from late-December to a peak in mid-January as the temperature cools. This change over time in density coincides with the observations of *Vidal & Rendón-Salinas (2014)*. *Vidal & Rendón-Salinas (2014)* suggested clusters of overwintering monarchs disaggregate when temperature increases, which is coherent with the temperature-density relation we report.

Our analyses of the *Brower et al. (2004)* samples for Conejos and Zapatero colonies led to considerably different conclusions than had we used the estimate they reported. If we had used their 50 million ha$^{-1}$ estimate as one of five (not six) samples in our analyses, we would have estimated a mixture distribution with a mean and median of 45.7 million butterflies ha$^{-1}$, respectively (code for analysis available in Code Supplemental Information 1). This estimate is twice our estimated median (21.1 million ha$^{-1}$). The reason

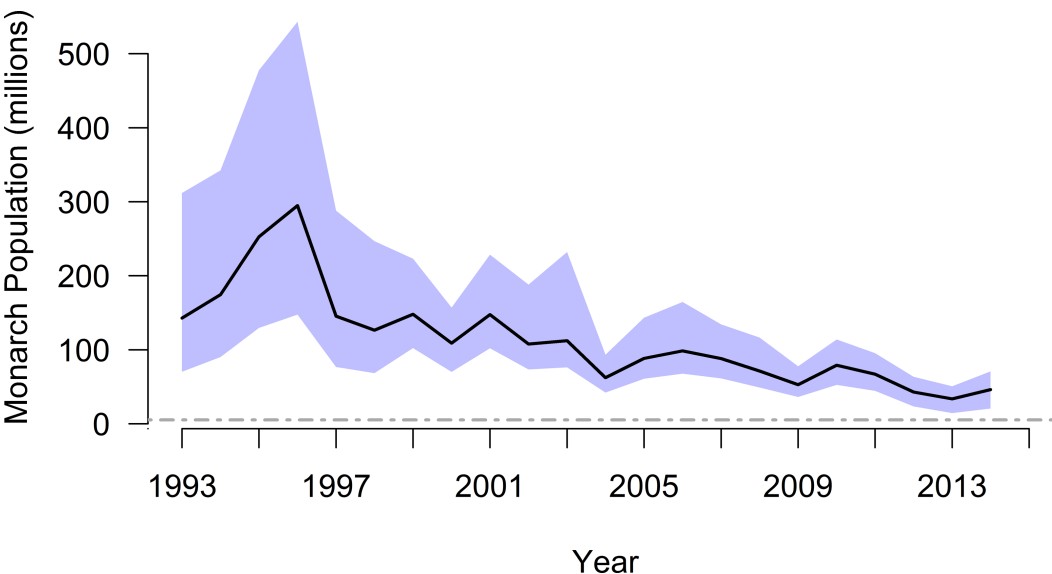

**Figure 4 Annual population size (with 95% CI), by year winter starts, for monarch butterflies over-wintering in Mexico.** The black line and associated blue confidence band depict patterns in annual abundance according to the full mixture distribution (i.e., mean density of 20.7 million ha⁻¹). The dashed gray line is an upper-end quasi-extinction risk threshold (0.25 ha) described by *Semmens et al. (2016)*.

*Brower et al. (2004)* concluded a density estimate of 50 million ha$^{-1}$ rather than the 15.9 and 18.3 million ha$^{-1}$ we report is because they used the mean to characterize the skewed distributions of their samples, a mean skewed to higher values by a few outlying estimates. Samples from both the Conejos and Zapatero colonies are small ($n = 29$) and skewed by a few very large counts (Appendix S1). For sufficiently large samples, the distribution of the sample mean is approximately normal according to the Central Limit Theorem. At the sample sizes reported by *Brower et al. (2004)*, we cannot be assured that the Central Limit Theorem holds. The preponderance of the observed data and the mass of the resulting distributions fitted to those data are considerably smaller than the mean (e.g., median for Conejos colony was 86 butterflies per sample versus a mean of 290 butterflies per sample). As a result, the values we drew from the fitted distributions for Conejos and Zapatero colonies led to a lower expected density than if we had used the higher published estimate.

The North American Climate, Clean Energy, and Environment Partnership Action Plan (*Trudeau, Obama & Nieto, 2016*) identified the restoration target of a 6-ha overwinter abundance of monarch butterflies as a goal to be achieved by Canada, the US, and Mexico by 2020. This 6-ha target is equivalent to approximately 127 million monarchs[1] (with a putative 95% confidence interval of 14–484 million) according to estimated median overwinter density. One of the central outstanding questions for conserving monarchs is, how much milkweed must be planted to create sufficient habitat to support this target population size? *Pleasants (2017)* suggested the loss of milkweed in the North Central region of the US between 1999 and 2014 amounted to 862 million stems, with an additional 1.34 billion stems remaining. *Pleasants (2017)* argued that milkweed in corn and soybean fields produced 3.9 times more monarch eggs than milkweeds in non-agricultural habitat (see

[1]Similar to a return to the 1998–1999 overwinter population, which was 6.02 ha and 137 million monarchs [71–267 million] (Table 2).

*Pleasants & Oberhauser, 2013*) and that, therefore, the loss of Midwestern agricultural fields was especially hard hitting to monarchs; he suggested that the loss of 850 million stems in corn and soy fields amounted to the equivalent of 3.31 billion non-agricultural stems of milkweed. Similarly, we estimate a 127 million monarch population would require 3.62 billion stems of milkweed; with 1.34 billion stems remaining in the landscape, the milkweed deficit could be as high as 2.28 billion stems, or ~700 million more stems than was needed according to *Pleasants (2017)* to return the population to 6 ha of occupied overwintering habitat. Alternatively, if we subtract the estimated amount of milkweed needed for a 126 million monarch population (i.e., 6 ha) from the equivalent needed for what *Pleasants (2017)* estimated remains (3.15 ha, or ~70 million monarchs), we obtain 1.8 billion stems (95% CI [0.7–4.70] billion stems), or 200 million stems more than estimated by *Pleasants (2017)*. These calculations assume a linear relationship between monarch and milkweed abundance but *Pleasants (2017)* demonstrated it was linear on the log-scale (i.e., Number of milkweed stems $= e^{0.12 \times \text{OW[ha]} + 7.24}$, where OW[ha] is number of hectares of butterflies overwintering in Mexico), indicating that our estimates of the milkweed deficit provide liberal upper bounds on what may be required.

Clearly, despite our best efforts at synthesizing the available information pertaining to overwinter density, there remains considerable uncertainty in the estimated densities, which in turn influences uncertainty in subsequent calculations of population size and associated levels of milkweed needed to sustain the species. Mixture distributions are often developed when data are believed to arise from more than one generation process or physical mechanism. Densities of overwintering monarchs reported by *Calvert (2004)* and *Brower et al. (2004)* may have differed for climatic, seasonal, behavioral, population size, or habitat-related reasons. Our climate regression suggested approximately half of the variation in density was attributable to variation in temperature-related climate variation. Remaining variation is likely to be explained by other factors. For instance, Calvert's studies were conducted in 1979 and 1985–1986, whereas Brower et al.'s data were collected in 2002. Similarly, Calvert collected data from the El Picacho and Sierra Chincua colonies, whereas Brower et al.'s findings came from the Zapatero (also known as Sierra Chincua) and Los Conejos (also known as El Rosario) colonies; these colonies occur across the Corredor Chincua-Campanario-Chivati-Huacal (*Garcia-Serrano, Reye & Alvarez, 2004*; Fig. 2 in *Slayback et al., 2007*), possibly contributing to environmental variation in density. Further, the methods used to generate these density estimates differed substantially, ranging from capture-mark-recapture methods to extrapolations based upon mortality estimates and structural characteristics of Oyamel fir trees.

At this time, we have too little information to posit an advantage of one density estimation method over another, but it is likely that a combination, nay, a mixture, of reasons contributed to differences in estimates. As a result, the best we can do is acknowledge the extent of uncertainty in our estimates. The mixture distribution we derived provides a reasonable articulation of the uncertainty associated with overwinter density estimates. The magnitude of uncertainty in the estimated density suggests the mean density is known within no better than a range of 1/3 to 3 times the expected value. Our estimates of uncertainty, however, may change with changes in assumptions. If we had

assumed the branch-based estimate of *Calvert (2004)* was correct within a factor of 3, 4 or more, for instance, the uncertainty in this estimate would have also contributed additional uncertainty in the final estimates. We assumed density might change both within and among years; stochastic variation around an unvarying mean density would not alter our conclusions, however. Systematic change in mean density, though, especially as population size declines, could have serious consequences on our inferences. Longtime observers of the overwintering colony (K Oberhauser, pers. obs., 2016 and L Brower, pers. comm., 2016) have suggested in recent years that monarchs are less densely packed on trees at the edge of the colony compared to trees in the center. Smaller colonies with a higher ratio of edge to inner trees could lead to uniformly less dense colonies which, in turn, would result in a systematic decline in density with population size and thus invalidate our application of density estimates to the current population; as a result, the observed decline in abundance would be even steeper than we report. These various assumptions point to the need for increased understanding of factors contributing to variation in overwinter density.

Precision of overwinter density estimates can be improved by measuring natural variability in species response to environmental conditions over time and space, focusing on robust parameter measurement and estimation error, and examining assumptions in models or extrapolations of these models. Capture-mark-recapture methods such as those employed by *Calvert (2004)*, but replicated over years and locations, seem to offer the most promising means of accomplishing this goal of improved overwinter density estimation. Capture-mark-recapture is the most common method for estimating population size in butterflies (e.g., *Gall, 1984*; *Bergman, 2001*; *Baguette & Schtickzelle, 2003*; *Haddad et al., 2008*), and its systematic use in the high-elevation Oyamel fir forests of central Mexico would enable robust estimates of daily and total overwintering population sizes, as well as survival and emigration probability (*Williams, Nichols & Conroy, 2002*). However, capture-mark-recapture methods come at the considerable cost of disturbing overwintering individuals, a practice that is currently disallowed and arguably not prudent given the small population size and the negative impacts of disturbance. Increased disturbance of overwintering individuals quickens fat depletion, disrupts the thermal advantages of communal roosting, and exposes the butterflies to predation and colder temperatures if they are unable to fly back into trees. The structural extrapolations employed by *Calvert (2004)* also result in considerable disturbance. Thus, while estimating density during the late-December reporting period for World Wildlife Fund-Mexico may be essential for understanding how many monarchs to attribute to the area over which they occur, accurate measurements using traditional approaches may come at considerable risk to the butterflies. Less invasive possibilities for estimating density, such as colorimetric analysis of the intensity and area of occupied trees (*Williams & Brower, 2016*), may prove to be a rewarding alternative.

## CONCLUSIONS

Combining the results of several studies conducted between 1979 to 2002, we conclude an estimate of 21.1 million butterflies ha$^{-1}$ is the most meaningful value when translating area occupied by overwintering monarchs into estimates of population size. While this

represents the best of our knowledge to date, the number of studies estimating densities of overwintering monarchs is small, and large discrepancies exist among various estimates, leading to considerable uncertainty. A better understanding of the spatial and temporal factors influencing monarch densities in their overwintering colonies is needed to accurately understand monarch population size, population viability, and characteristics of the environment required for sustaining the species at desired levels of abundance. However, we acknowledge that this information may be difficult to attain, and that continued careful monitoring of area occupied and non-intrusive estimation of relative density, along with an understanding of the degree to which habitat restoration (and degradation, *Brower et al., 2016*) is occurring, may provide our best understanding of the critical relationship between milkweed availability and monarch numbers.

## ACKNOWLEDGEMENTS

We thank L Brower for data used in our analyses. We thank A Agrawal, L Brower, A Shapiro, and an anonymous referee for comments greatly improving the quality of this contribution. Any use of trade, product, or firm names are for descriptive purposes only and do not imply endorsement by the US Government. The views expressed in this article are the authors' own and do not necessarily represent the views of the US Fish and Wildlife Service.

### Funding

This work was supported by the John Wesley Powell Center for Analysis and Synthesis of the United States Geological Survey. The funders had no role in study design, data collection and analysis, decision to publish, or preparation of the manuscript.

### Grant Disclosures

The following grant information was disclosed by the authors:
John Wesley Powell Center for Analysis and Synthesis of the United States Geological Survey.

### Competing Interests

The authors declare there are no competing interests.

### Author Contributions

- Wayne E. Thogmartin conceived and designed the experiments, performed the experiments, analyzed the data, contributed reagents/materials/analysis tools, wrote the paper, prepared figures and/or tables, reviewed drafts of the paper.
- Jay E. Diffendorfer, Karen Oberhauser, Darius Semmens and Ruscena Wiederholt conceived and designed the experiments, wrote the paper, reviewed drafts of the paper.
- Laura López-Hoffman and Orley R. Taylor conceived and designed the experiments, reviewed drafts of the paper.

- John Pleasants conceived and designed the experiments.
- Brice X. Semmens conceived and designed the experiments, contributed reagents/materials/analysis tools.

## Data Availability

The raw data has been supplied as elements within R statistical code, which is included as a Supplementary File.

## Supplemental Information

Supplemental information for this article can be found online at http://dx.doi.org/10.7717/peerj.3221#supplemental-information.

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
