# Peer review of "Density estimates of monarch butterflies overwintering in central Mexico"

_PeerJ, doi:10.7717/peerj.3221_

## Round 0.1 · original submission · Major Revisions

Please revise your paper, taking the comments of the reviewers into account. Please take extra care to rebut the criticisms of the most critical reviewer. Note that your paper will be sent out to review again, and thus it is essential to address all of the reviewers' comments.

·

Basic reporting

No comments.

Experimental design

The statistical approach taken here is highly innovative. The innovation is necessary given the rather extreme heterogeneity of the data and methods being used in the meta-analysis. It is thus necessary that the methods used in this paper be presented very explicitly, allowing for replicability and potential comparisons with other approaches. I find that this has been done admirably and particularly in MS lines 279-295.

Validity of the findings

The authors have been very explicit in presenting a suite of caveats concerning the validity and generalizability of their conclusions (MS line 317 ff.). This is especially important given the potential attribution of greater authority to those conclusions in management decision-making than is warranted, given the very high level of uncertainty. As a reviewer, I am somewhat handicapped by not having access to "Pleasants, in press," which is to say I am constrained from commenting on the robustness of estimates of numbers of milkweed stems available and/or necessary to meet management goals.

Additional comments

This is a rather general comment: Estimates of milkweed needs are cast in some doubt by the issue of context. Why should milkweed in ag fields be so heavily used, as against milkweed elsewhere? If the contextual determinants of oviposition behavior can be better understood, utilization of planted (substituted) milkweed resources might be maximized; it is entirely possible that due to context, much of the planted (substituted) milkweed might be little utilized by ovipositing females. This just adds one more layer of uncertainty to a project that already has many such layers! You might add something to this effect as just one more in your string of caveats.

·

Basic reporting

I have read the manuscript and critiqued several relevant aspects of the biology of overwintering butterflies in Mexico that bear on the problem of determining the most reliable way of estimating the numbers of butterflies in the overwintering colonies.
My mathematical prowess is not high enough to judge whether there are any problems with the statistical models...so I suggest that the authors take into consideration my comments and suggestions (and perhaps those of the second reviewer), revise accordingly, and re-review.

I will attach or paste in the msw version of the ms that I have read and made comments and suggestions in orange.

The purpose of the ms is to review the colony estimate data to delimit the strong variability in numbers which are expressed as a function of the number of hectares occupied by all known colonies of butterflies. Wide variance characterizes the sampling efforts making it difficult to make realistic recommendations for mitigation. This is important to the current efforts to promote the recovery of the eastern North American butterfly population by planting a realistic number of milkweed foodplants to restore those lost due mainly to industrialized agriculture, herbicide and genetically resistant corn and soybean crops. The paper is straight forward and well presented, but I see a two principal problems.
1, There is an error in the statement re the overall differences in the storm mortality data sets for Conejos and Zapatero. . Lines 201-205. Statement in italics in the included draft of LPB critique of original ms. is wrong. Approximately half of butterflies from one colony, Conejos, were deemed dead or moribund ((actual data Conejos - 99.9% dead; Zapatero. 98.6% dead)) whereas all of the butterflies at the Zapateros colony were dead or moribund; thus, to calculate density for the Conejos colony, counts were doubled. SO this is a major error.. So the % mortality was NOT NOT different. So the model should probably be rerun WITHOUT doubling the Conejos data,
2. My other major concern is that the conversion of hectares to numbers of butterflies makes the assumption that the density of butterflies on the trees and in the colonies has remained constant over the years. Several of my colleagues and I have noted in the last few years that the density of butterflies in the clusters is far less than in the past years. Unfortunately there are no data on this. The problem with this issue is that if the colony densities diminish, the hectare approach inflates the numbers by focusing only on area measurements. Please see several comments and suggestions in orange in the attached copy of the ms, In one of the Tables Zapateros should say Zapatero. I suggested adding two references, and there is a remark about the Conejos and Zapatero original data which should be attributed to LPB with a citation.

Experimental design

See above

Validity of the findings

See above.

Additional comments

See above

Reviewer 3 ·

Basic reporting

This paper describes an attempt to derive an estimate of the density of monarch butterflies, from the eastern North American population, that overwinter in Mexico each year. Apparently, the monarchs that overwinter in Mexico tend to cluster tightly in trees and are quite dense on the branches, so that the measures of overwintering size used have always been a measure of colony ‘area’. This is a very crude estimate, but it nevertheless has allowed researchers to track changes in colony ‘size’ over time. The authors of this manuscript sought to create an approximation for how this area measurement translates into actual numbers of monarchs – i.e. numbers per unit area. This goal seems worthwhile from a management standpoint. Unfortunately, this manuscript suffers from a number of serious issues, which makes this unworthy of publication. I list these issues below.

1. The idea of estimating abundance of monarchs at this overwintering site is laudable, but the approach used here was to take a very small number of previously-published estimates and come up with a very fancy ‘average’ value of those estimates. Thus, the dataset used in this effort, if you can call it that, is a set of 6 numbers, I believe. It was actually listed as both 6 and 5 in various places, so it is not clear what the final number was. Either way this is a ridiculously-low number to base an entire research study on, regardless of how much the authors have dressed-up their analyses of those data. Moreover, 4 of the estimates came from a single publication!

2. Regardless of how few data there were, there is some question as to their accuracy in the first place. For starters, why do the previous estimates of density/area differ so much? These 6 numbers seem to vary greatly, which makes me question if deriving an average of all of these values would even be appropriate in the first place. The variation from study to study in the estimates could be a sign that these monarchs actually vary from year to year in their clustering density. Moreover, when the estimates of colony area are announced each year, I note that there is never an announcement of how many trees are occupied that year. If in one year there are monarchs covering 5 hectares, and there are 25 trees covered that year, that would be greatly different than a year when the colony covers 5 hectares but only 15 trees within that area. Also of note is that none of the published estimates come from recent years – they are from the late 70s to the early 2000s. How do we know the clustering density from the early years is the same as it is now?

3. There is a section in this manuscript that attempts to relate the number of monarchs counted in Mexico to the number of milkweeds in the breeding range (milkweed being the larval hostplant of monarchs). Given the stated goal of this project was to estimate monarch clustering density in Mexico, this section did not seem appropriate here, and in any case, it is just wrong anyway – earlier this year I read a paper from a group from Cornell (Inamine and colleagues), who examined this whole idea that monarchs are losing milkweed, and that is what is causing the declines in Mexico. Surprisingly their analyses showed milkweed is not limiting to monarchs, and that the overwintering declines in monarch abundance that have everyone worked up over, are not happening during the summer. That project, and the data it relied on, seemed pretty convincing to me. But for some reason this very influential paper was not even mentioned in this manuscript. Not only that, but the authors here have attempted to calculate how many milkweeds are needed to grow the winter population, when in fact milkweed seems to have little to do with the numbers in Mexico. So the exclusion of such an influential study from this paper can only be interpreted as either a glaring oversight by these authors, or a veiled attempt to marginalize those findings, perhaps because they run counter to prior work conducted by these authors. I suspect it is the second of these scenarios, but either way it looks bad for these researchers.

4. There is another oddly-placed section in this paper, which is the attempt to relate clustering density to environmental conditions. Again, this entire analyses is based on 5 or 6 data points, spanning many years. I think the authors pooled them, looked at the calendar dates of the estimates, and found corresponding environmental data from those dates. Then they looked to see if the density estimate varied with the environmental conditions – I think. Either way, this is rubbish. Throughout this paper the authors spent a great deal of time describing how each of these estimates was conducted in a different way, and in different years, different locations, etc., so then it is ridiculous to think that the variation in these numbers has any meaningful bearing on environmental conditions.

Experimental design

See comments above

Validity of the findings

See comments above

Additional comments

On a final note, as someone who works on other insect systems, I find the attention given to the monarch and its conservation to be grossly misplaced, especially given the preponderance of evidence that this species itself is in no danger of extinction. Meanwhile, there are other invertebrate species around the world and in the U.S. that are in real jeopardy, and that aren’t getting any attention at all. It seems to me that a lot of the hype over the monarch is being caused by alarmist reports and publications just like this one, which appears to be designed to manufacture media headlines. In the case of this study (if it were ever published) the headline would read: “Scientists discover fewer monarch butterflies than once thought” I say this, not out of bitterness, but to urge these authors to take a less dramatized tone in the future writings. For example, based on the recent Inamine et al study (which was not cited here), introductory statements like "The eastern population of monarchs has declined by 90%" are not correct, or at the very least, misleading - it is more precise to say "The cohort of individuals that reaches Mexico has declined by 90%, though the abundance of breeding monarchs has not changed."

---

## Round 0.2 · Minor Revisions

Thanks so much for your careful responses and revisions. Please deal with the few remaining concerns of reviewer 2, which you should be able to do promptly.

·

Basic reporting

No comment

Experimental design

No comment

Validity of the findings

No comment

Additional comments

I read the first set of reviews and judged the long critical one to be a largely unjustified diatribe. The rest of the criticisms can likely be addressed by the authors.

I think this paper is important to publish so that the various conservation approaches and groups are all working from the same set of assumptions re (1) the numbers of milkweeds needed to assure a minimal 6 ha of overwintering area and (2) that the median density is currently accepted as 21 million monarchs per hectare at the Mexican overwintering sites.

While I am not well enough versed to critique the mathematical models used in the paper, I think the conclusions reached are as reasonable as is currently possible without further research in assessing numbers per unit area in Mexico.

My most serious concern as I have expressed is that the density of butterflies in the clusters is likely getting lower each year and that sticking with 21 million per ha will or is overestimating the numbers. On the other hand, the estimate is a carefully reasoned one and the best we have at the present critical time when milkweeds need to be replaced. At the worst, too few milkweeds may be planted, certainly not too many.

I strongly and heartily recommend that the paper be published as soon as possible but with a clearer caveat on the likely density problem.


Specific editing comments:
line 45. Change "is needed" to "are needed"

Line 48 sentence is not clear re what other published estimates is referring to.........and ambiguous re butterflies or milkweeds.

Lines 63-64. ..." Clustering in densely packed colonies on the lower branches of Oyamel fir trees also minimizes mortality...." While the clusters provide some microclimatic protection, the main benefits to the clusters are the blanket and umbrella effect provided by the intact forest...the forest cover reduces wetting and radiant heat loss.

Lines 67-68. Subtle correction. " In early spring, the annual migration of monarchs over much of eastern North America commences from this location,” The annual migration begins in the fall, not the spring. These fall butterflies overwinter and the survivors (the same generation) remigrate north to the Gulf Coast area. The way the sentence is written implies that the spring remigration involves a fresh new generation.

Line 81. Recent is redundant in sentence

Line 83.. (in the winter of 1996)

line 89. I think it important to add: "and forest degradation in the Mexican overwintering habitat (Brower et al. 2016). Our recent paper documents ongoing deterioration and the recent salvage logging of a purported 24,000 trees from the heart of the Reserve is certain to be detrimental. You cite our paper elsewhere in the ms so it is not adding a reference.

line 97.. after monarch butterflies insert in the Mexican overwintering sites.....

line 115-121. Calvert estimates: text says 25 years later when the actual counts were made for the branch and trunk counts in the 78-79 overwintering season and for the mrr study in the 85-86 ow season (all published in 2004).. You give this later but it is a little misleading here,

lines 132 - 135. Sentence is vague, Need to be clearer and especially to make a more definitive statement re diminishing cluster density through time . I see you get to this at the end of the paper but it is important enough that maybe refer to it here?

line 173. not 1977. Jan 1979

line 221. I do not understand how dew point is relevant to the overwintering. Could authors add a couple of sentences explaining /?

line 325-327. Densities of overwintering monarchs reported by Calvert (2004) and Brower et al., (2004) likely differed for climatic, seasonal, behavioral, population size, or habitat-related reasons. change likely to could have. Also, does population size mean colony size when measured?

line 332- 335. Brower et al.’s findings came from the Zapatero (also known as Sierra Chincua) and Los Conejos (also known as El Rosario) colonies; these colonies occur across the Corridor Chincua Campanario-Chivati-Huacal (Garcia-Serrano et al., 2004; fig. 2 in Slayback et al., 2007), possibly contributing to environmental variation in density. My sense of history on this is that all the colonies in the early days were remarkably uniform in how the trees were densely covered and differed principally in the area of the colonies

lines 350-358 Systematic change in mean density, though, especially as population size declines, could have serious consequences on our inferences. Longtime observers of the overwintering colony (K. Oberhauser and L. Brower, personal communication) have suggested in recent years that monarchs are less densely packed on trees at the edge of the colony compared to trees in the center. Smaller colonies with a higher ratio of edge to inner trees would result in a systematic decline in density with population size and thus invalidate our application of density estimates to the current population; as a result, the observed decline in abundance would be an even steeper than we report. Karen and Lincoln likely disagree on edge vs. colony inner density. My sense is that the entire colony density is now uniformly thinner than it was years ago and is becoming thinner each year. I would make this possibility clear and advocate that a methodology to measure density is urgently needed because if true, the hectare measurements are becoming increasing overetimates of the actual numbers.

lines 359-378. The difficulty of carrying out MRR studies and their disruption of the colonies is a problem. I see two solutions: 1) Convince the Reserve Director that one area be set aside for research only and carry out MRR yearly excluding the public. 2) The other is as Williams and Brower suggested is to develop drone photography followed by colorimetric analysis of the intensity and area or orange coloration as a repeatable and reliable measure of density. There is a colony outside the Reserve called Palomas that might be the ideal one to use. I note that the authors suggest Lidar. Maybe all three measures could be done.

---

## Round 0.3 · accepted · Accept

Thank you very much for your prompt and thorough revisions.